# Large household reduces dementia mortality: A cross-sectional data analysis of 183 populations

**Wenpeng You[1]***, **Maciej Henneberg[1,2]**

**1** School of Biomedicine, The University of Adelaide, Adelaide, Australia, **2** Institute of Evolutionary Medicine, University of Zurich, Zürich, Switzerland

\* wenpeng.you@adelaide.edu.au

## Abstract

### Background

Large households/families may create more happiness and offer more comprehensive healthcare among the members. We correlated household size to dementia mortality rate at population level for analysing its protecting role against dementia mortality.

### Methods

This is a retrospective cross-sectional study. Dementia specific mortality rates of the 183 member states of World Health Organization were calculated and matched with the respective country data on household size, Gross Domestic Product (GDP), urban population and ageing. Scatter plots were produced to explore and visualize the correlations between household size and dementia mortality rates. Pearson's and nonparametric correlations were used to evaluate the strength and direction of the associations between household size and all other variables. Partial correlation of Pearson's approach was used to identify that household size protects against dementia regardless of the competing effects from ageing, GDP and urbanization. Multiple regression was used to identify significant predictors of dementia mortality.

### Results

Household size was in a negative and moderately strong correlation (r = -0.6034, p < 0.001) with dementia mortality. This relationship was confirmed in both Pearson r (r = - 0.524, p<0.001) and nonparametric (rho = -0.579, p < 0.001) analyses. When we controlled for the contribution of ageing, socio-economic status and urban lifestyle in partial correlation analysis, large household was still in inverse and significant correlation to dementia mortality (r = −0.331, p <0.001). This suggested that, statistically, large household protect against dementia mortality regardless of the contributing effects of ageing, socio-economic status and urban lifestyle. Stepwise multiple regression analysis selected large household as the variable having the greatest contribution to dementia mortality with R$^2$ = 0.263 while ageing was

**Data Availability Statement:** All the data supporting our findings in this paper were freely downloaded from the UN agencies' websites. The data sources have been described in Section 2.2 Data Sources, and their sources have been

referenced in the text. The whole set of the data for this study has been attached as well. The title of the attached data is S1 Data: S1 A whole set of data for this study. No ethical approval or written informed consent for participation was required.

**Funding:** The authors received no specific funding for this work.

**Competing interests:** The authors have declared that no competing interests exist.

**Abbreviations:** GDP, Gross Domestic Product; Life e$_{(60)}$, Life expectancy at 60 years old; SES, Socioeconomic status; UN, United Nations; WHO, World Health Organization; DMR, Dementia mortality rate.

placed second increasing $R^2$ to 0.259. GDP and urbanization were removed as having no statistically significant influence on dementia mortality.

## Conclusions

While acknowledging ageing, urban lifestyle and greater GDP associated with dementia mortality, this study suggested that, at population level, household size was another risk factor for dementia mortality. As part of dementia prevention, healthcare practitioners should encourage people to increase their positive interactions with persons from their neighbourhood or other fields where large household/family size is hard to achieve.

## 1. Introduction

Dementia is an umbrella neurological syndrome resulting from more than 100 brain disorders, most common of which include Alzheimer's disease, vascular diseases, Lewy bodies, frontotemporal disease etc. [1, 2]. Worldwide, it is estimated that 5–8% of general population aged 60 and over have been diagnosed with dementia in past years [2]. It has become one of the most common causes of dependency and disability among the elderly [1, 2].

Dementia produces an increasing societal burden resulting in a total economic costs of US\$ 818 billion (1.1% of the world's gross domestic product) in 2016 [2], which has been considered as the major challenging health issue in 21$^{st}$ century [3, 4]. Many health professionals do not follow the regulations or facility policies about human rights and freedom, and still consider physical or chemical restraints as the inevitable approach to control patients' behavioural symptoms and prevent the disruption of life-sustaining therapies [2]. Furthermore, dementia patients and their families are subject to prejudice, and their life quality is affected [1].

Since 1906 when Alzheimer's disease has been associated with dementia, enormous investment has been allocated for studying dementia prevention and treatment [5, 6]. Ageing and family history are associated with genetic background which are easily recognized as the irreversible risk factors for dementia [2, 7, 8]. The environmental and behavioural risk factors, such as sedentary lifestyle, lower socioeconomic status (SES) and unhealthy diet have been circumstantially postulated as the risk factors for dementia [7, 9–12]. Studies have consistently revealed that negative psychological functioning, such as depressive symptoms [13] and neuroticism [14] are the risk factors for dementia development. However, to date, the aetiology and pathology of dementia are still not well understood and there is still no treatment currently available to cure dementia or to reverse its progressive course [2]. A statistical study of the prevalence of dementia, or its ultimate result–mortality, globally and in countries grouped by their socioeconomic characteristics may shed light on the role environmental factors play in aetiology of this condition.

Human species had lived in small hunter-gatherer groups for millions of years before they started to live in large scale societies some ten thousand years ago [15]. During the hunting and gathering period, humans have well adapted to cooperative breeding [16, 17], and then evolved alloparental care [18]. Therefore, human's millions of years of adaptation suggest that biological foundations of human love have genetically shaped humans for flourishing in small communities [19, 20]. However, in the last few hundreds of years, human societies were industrialized quickly. The rapid industrialization has made most people grow up in just core families with few siblings, which is different from how humans had adapted for flourishing. Such

discrepancies or mismatches have been associated with mental health in human population [21, 22].

It is well researched that positive psychological well-being has been implicated in health across adulthood [23]. Household creates a social environment which is salient to maintain health for the co-residential members. On a daily basis, the individual members encounter this environment, play their social role and enjoy the social relations [24]. Moreover, studies also showed that large household offers the residents the subjective happiness [25] leading to low risk for residents to develop various cancers [26], for instance female breast cancer [27, 28] and ovarian cancer [29]. Subjective happiness was associated with mental health significantly stronger than with physical health in people with disabilities [30] and hospital patients [31]. A recent study revealed that greater household size has the protecting role against children developing mental health disorders [32].

The above considerations directed us to try to identify whether smaller household could serve as a risk factor for people to develop and die of dementia. Therefore, in this study, we assessed, from a global perspective, whether large household has the inhibitory role in lowering the risk for the residents to die of dementia using empirical population level data obtained from international organizations. This approach has been already used by our team in a number of studies on various other aspects of human health [26, 28, 29, 33–40].

## 2. Materials and methods

### 2.1 Study design

This is a retrospective cross-sectional study using data already reported by international organisations. In epidemiology this type of study is considered an "ecological analysis"

### 2.2 Data sources

The population level data were collected for this ecological study. A whole set of data was attached (S1 Data).

1. Population specific dementia mortality rate (DMR, per 100,000) was calculated as the dependent variable.

The comprehensive and comparable assessment of country specific number of deaths due to "Alzheimer disease and other dementias (GHE Code 950, Line 140)" and the country specific total population (ISO-3 Code, Line 10) were provided by the WHO Global Health Estimates [41, 42].

DMR estimates are integrated by the team of WHO Global Health Estimates considering the latest available mortality and cause distributions reported by each country and most recent information from various WHO programs for causes of public health importance. This increases the comparability of DMR between countries. WHO website published the description of data, methods and cause categories in a Technical Paper [43]. Table 4.1 and figures 1.1, 4.1, 9.1 in this document explain in detail how the number of deaths due to Alzheimer disease and other dementias estimates were obtained, while its Section 10 refers to quantitative uncertainty ranges that are available as part of the comprehensive GHE 2016 estimates on the WHO website. Though it is difficult to repeat here all careful considerations of the WHO team involved in obtaining as reliable as possible estimates it may be mentioned that no estimates were provided for WHO member countries whose populations were less than 90,000 in 2016. Thus, data for these countries are missing. Obviously, the estimates are not a 100% reliable data, but they are the best information available. The formula for dementia mortality

calculation is below:

$$Dementia\ mortality\ rate$$
$$= \frac{Total\ number\ of\ persons\ who\ died\ of\ Alzheimer\ and\ other\ dementias}{Total\ population} X\ 100,000$$

The United Nations and its agencies define the countries and territories in different ways. In order to avoid this conflict, both the country and territory are called "population" in the study.

2. The population specific household size was extracted from the United Nations Booklet as the predicting/independent variable [44]. In this document data sources are described as follows (Page 31): "*The database comprises estimates of household size and composition obtained through analysis of microdata from the following data sources: Demographic and Health Surveys (dhsprogram.com), European Union Labour Force Surveys (http://ec.europa.eu/eurostat/web/microdata/european-union-labour-force-survey) and microdata samples maintained by Minnesota Population Center, Integrated Public Use Microdata Series, International: Version 6.4 [dataset]. Minneapolis, MN: University of Minnesota, 2015. http://doi.org/10.18128/D020.V6.4. Selected estimates of the average household size and headship rates were obtained through secondary sources, including the Demographic Yearbook of the United Nations (https://unstats.un.org/unsd/demographic/ products/dyb/dyb_Household/dyb_household.htm) and published reports of censuses.*"

The household is a fundamental socio-economic unit in human societies. It consists of one individual or a group of people, regardless of whether there are any kinship ties, living together for sharing food, shelter and other daily life essentials. Therefore, household refers to people living together in a housing unit who may or may not be family members.

Household relations are usually characterized with family relationships because they are invested with the powerful norms, histories, and emotions which originated from family [23, 45–47]. Therefore, in this study family and household are used interchangeably.

Considering the majority of dementia patients are cared for at home which is called "informal" care, household size may represent the level of care which the patients can receive.

3. The WHO published life expectancy at 60 (Life $e_{(60)}$) was selected as a potential confounder [48].

Ageing has been a well-known significant risk predictor of dementia. Most of the studies include 65 years age as the start of ageing for reporting the prevalence and incidence of dementia. However, the United Nations generally use 60+ years to refer to the older population, and it takes years for dementia associated symptoms and signs to appear because development of most types of dementia is slow and progressive [49, 50]. Therefore, in this study, Life expectancy at age 60 years ($e_{(60)}$) was considered as the indicator of ageing.

4. The World Bank published data on Gross Domestic Product (GDP) and urbanization were also included as the potential confounder [51]. The World Bank website provides detailed information explaining the validity and reliability of such data provided (http://opendatatoolkit.worldbank.org/en/supply.html in its section of Supply and Quality of Data).

GDP was expressed in per capita purchasing power parity (PPP in current international $) in 2010. SES has been associated with prevalence and mortality rates of dementia, and also with regional variation of dementia prevalence. GDP PPP was included as the confounding factor because it relates to the levels of healthcare service which affects the mortality rate. GDP may also affect quality of information reported to international organisations.

Urbanization was measured with the percentage of total population living in urban communities in 2010. Urbanization represents the demographic trend in which more and more

population has become concentrated in urban communities. It entails air pollution, consumption of food with few nutritional benefits, but energy dense, high levels of salt, fat, sugar and alcohol. Urbanization is also associated with less physical exercise, obesity and overweight. Therefore, urban living has been considered as a complex risk factor for chronic diseases [52].

The international organizations, the United Nations, WHO and the World Bank have monitored and published the data on the population specific economic status, health risk factors and nutrition and diet intake for decades. These data have been assisting government policymakers and funders to track and investigate priorities of health research and development based on public health needs, while ensuring that funds and resources are used to meet the priorities. Their data have been more and more used in academic area for identifying the relationships between population level health and their risk factors. For example, recently they were used to examine the relationships between nutrients and obesity [35, 53–55], diabetes [33, 56–58], and the relationship between relaxed natural selection and obesity [36, 37]; type 1 diabetes [33] and cancers [28, 30, 36] respectively.

## 2.3 Data selection

In order to capture as many populations as we could to increase the sample size for this study, we kept the full list of 183 WHO member states (populations) which have the data available for dementia mortality rate calculations. Details of the way these data were produced are described in the WHO Technical Paper [43]. This paper specifies that: "... *most recent vital registration (VR) data for all countries submitting VR data to the WHO Mortality Database (WHO MDB), where the VR data meets certain criteria for completeness and quality;* ... "*were used together with careful estimates for other countries. [43]*.

Greater household size not only shows its beneficial effects on protecting residents from developing dementia, but also offers its supporting or caring role in reducing the risk for patient to die from dementia. The dementia mortality rate could manifest the level of beneficial effect for dementia patient during their whole life span.

Population specific household size, urbanization life expectancy and GDP PPP were matched for those states with dementia mortality rate data. We obtained most recent population specific dementia mortality rates (N = 183) through calculation, household size (N = 170), GDP PPP (N = 178), Urbanization (N = 183) and ageing (N = 183) through extraction. Each population was considered as an individual research subject in the analysis. Therefore, numbers of populations included in analyses of relationships between variables may differ somewhat because all information was not uniformly available for all states.

All the aforementioned data were freely available from the websites of the UN agencies.

## 2.4 Data analysis

Scatter plots were produced in Excel (Microsoft® 2016) to explore and visualize the correlations between household size and dementia mortality rates. Scatter plots also allowed us to assess data quality and distributions of the variables.

Prior to correlation/regression analyses all data were log-transformed (ln) in order to reduce non-homoscedasticity of their distributions and possible curvilinearity of regressions. To assess the relationships between household size and dementia mortality rate in different data analysis models, the analysis proceeded in four (4) steps.

1. Pearson's and nonparametric correlations (Spearman's rho) were used to evaluate the strength and direction of the associations between household size and all other variables, including independent variables and competing variables.

2. Partial correlation of Pearson's moment-product approach was used to assess the relationship between household size and dementia mortality rate while we controlled for ageing, GDP PPP and urbanization which have been commonly considered as the contributing factors of dementia.

We alternated the four variables (DMR, ageing, GDP PPP and urbanization) as the independent predictor to explore its relationship to DMR while keeping all the other three variables statistically constant. This allowed us to analyse and compare the levels of correlations between DMR and four potential risk factor while controlling for the other three variables [28, 39]. Subsequently, we alternately controlled for each variable as the potential confounder for analysing if and how much it could explain the correlation between DMR and each of the three variables.

Fisher's r-to-z transformation was performed to test significance of differences between correlation coefficients.

3. Standard multiple linear regression (enter and stepwise) was performed to visualize the relation between DMR and each predicting factor and identify the most significant predictor (s) of DMR respectively. In order to explore if household size can partially explain why ageing, GDP PPP and urbanization were correlated with DMR, the multiple linear regression analyses were performed to determine the correlations between DMR incidence and the risk factors in two models, i.e. with and without incorporating household size as one of the predicting variables.

4. In order to demonstrate that correlation universally exists between household size and DMR regardless of these factors, populations were grouped for correlation analyses. The exploration into different correlations between household size and DMR also allowed us to compare the different levels of correlations in different country groupings. The criteria for grouping countries used the World Bank income classifications [59], WHO regions [60], countries sharing specific characteristics like geography, culture, development role or socio-economic status, like Asia Cooperation Dialogue (ACD) [61], Asia-Pacific Economic Cooperation (APEC) [62], the Arab World [62], Latin America and the Caribbean (LAC) [63], Southern African Development Community (SADC) [64] and Organization for Economic Co-operation and Development (OECD) [62]. All the population listings are sourced from their official websites for matching with the list of populations with DMR.

Pearson's, non-parametric Spearman's rho correlations, partial correlation and multiple linear regression (enter and stepwise) were computed with SPSS v. 27 (SPSS Inc., Chicago Il USA). The significance was reported when p-value was <0.05, but the significance levels of $p < 0.01$ and $p < 0.001$ were also reported. Regression analysis criteria were set at probability of F to enter $\leq 0.05$ and probability of F to remove $\geq 0.10$. The raw data were used for scatter plots in Excel® 2016.

## 3. Results

Fig 1 shows the relationship between household size and DMR. The relationship is negative along a power curve with moderately strong negative correlation (r = -0.6678, p < 0.001). Subsequent parametric analyses of log-transformed data and nonparametric analyses confirmed the relationship between DMR variables and the household size.

Table 1 presents relationships between all the variables (dependent and independent) in Pearson r (above the diagonal) and nonparametric (below the diagonal) analyses. Worldwide (n = 169), Spearman's rank correlation showed that household size was in significant negative correlation to DMR ($r = -0.579, p < 0.001$). This strength and direction of relationship were similar and observed in Pearson's $r$ household size and SMR variables (r = - 0.524, p<0.001).

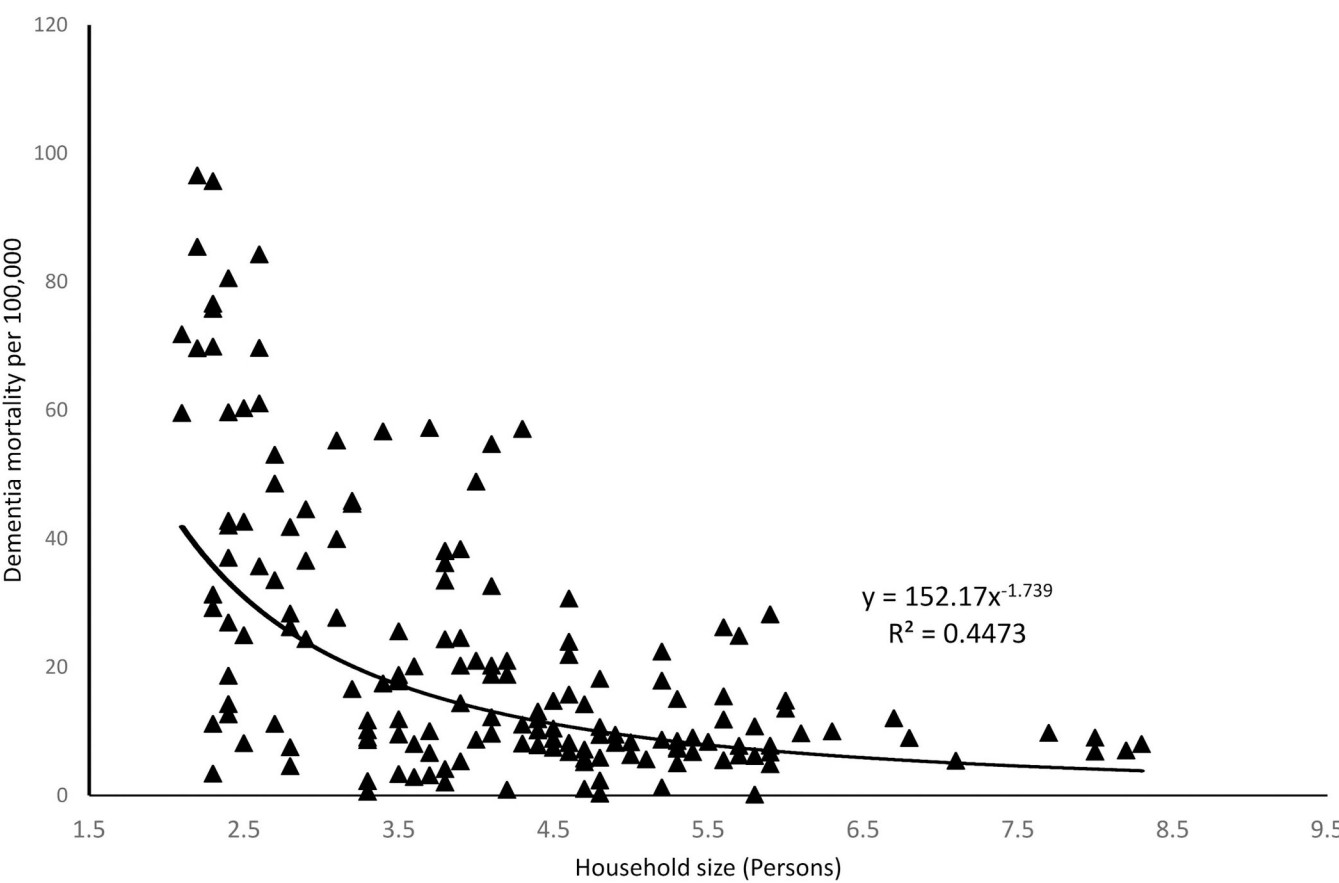

Data source & definition: Household size (the United Nations): the number of persons who make common provision of food, shelter and other essentials for living. Dementia Mortality Rate (World Health Organization): Calculated from Global Health Estimates. Both data were not log-transformed.

**Fig 1. The relationship between household size and dementia mortality rate.**

Worldwide, non-parametric analysis showed that DMR was associated with ageing (r = 0.533, p<0.001), GDP PPP (r = 0.497, p<0.001) and urbanization (r = 0.436, p<0.001). These strengths and directions of the relationship were observed in the Pearson analysis as well (Table 1).

Table 2 shows the relationship between DMR and household size, ageing, GDP PPP and urbanization examined by controlling for the other three variables in a partial correlation analysis. Household size was the only independent variable to have significant correlation (r = −0.332, p < 0.001) with DMR independent of the other three variables (Table 2). Neither GDP nor urbanization showed a correlation with DMR independent of the other three variables despite the fact that each of them (GDP and urbanization) had a significant correlation to DMR in simple bivariate analyses. This suggests that household size was the independent risk factor for DMR. This suggestion was proved true in the subsequent Stepwise linear regression analyses (Table 3).

Table 3 shows if and how much household size explained the correlations of the other three variables to DMR respectively. In the enter model, when household size was not considered as one of the independent variables, ageing was the only significant predictor of DMR (β = 0.357, p < 0.001). However, when household size was incorporated as an independent variable, it ranked as the strongest predictor of DMR (β = - 0.362, p < 0.001). Ageing was still a significant predictor of DMR (β = 0.226, p < 0.05), but the correlation strength has significantly decreased

**Table 1. Data descriptives and bivariate correlations (Pearson, above diagonal & non-parametric below diagonal) between all variables.**

|  | Household size | Dementia Mortality Rate (Per 100,000) | Ageing (Life expectancy at age 60 years) | GDP PPP (US$ (2010)) | Urbanization (Urban population percentage) |
|---|---|---|---|---|---|
| Average | 4.1 | 23.0 | 19.5 | 16,324.3 | 55.4 |
| Median | 3.9 | 12.2 | 19.0 | 9,715.0 | 55.5 |
| Standard deviation | 1.4 | 25.0 | 3.1 | 18,351.3 | 22.8 |
| Minimum | 2.1 | 0.1 | 13.0 | 646.9 | 10.6 |
| Maximum | 8.3 | 175.1 | 26.0 | 122,609.4 | 100.0 |
| CORRELATIONS |  |  |  |  |  |
| Household size | 1 | -0.524*** | -0.682*** | -0.628*** | -0.522*** |
| N | 169 | 169 | 169 | 165 | 169 |
| Dementia Mortality | -0.579*** | 1 | 0.458*** | 0.375*** | 0.318*** |
| N | 169 | 183 | 183 | 177 | 183 |
| Ageing | -0.689*** | 0.533*** | 1 | 0.760*** | 0.592*** |
| N | 169 | 183 | 183 | 177 | 183 |
| GDP PPP | -0.648*** | 0.497*** | 0.760*** | 1 | 0.747*** |
| N | 165 | 177 | 177 | 177 | 177 |
| Urbanization | -0.564*** | 0.436*** | 0.650*** | 0.791*** | 1 |
| N | 169 | 183 | 183 | 177 | 183 |

Significance level: * p<0.05

** p< 0.01

***p< 0.001.

Data sources & definitions: Household size (the United Nations): the number of persons who make common provision of food, shelter and other essentials for living. Dementia Mortality Rate (World Health Organization): Calculated from Global Health Estimates. Ageing (the United Nations) measured with the Life Expectancy at 60 years old. Per capita GDP PPP (the World Bank): the per capita purchasing power parity (PPP) value of all final goods and services produced within a country in a given year. Urbanization (the World Bank): the percentage of population living in urban area.

(z = 1.67, p<0.05). Both GDP and urbanization barely showed correlations with DMR regardless of household size inclusion.

Table 3 also shows that when household size was not included as one of the independent variables, ageing ($R^2 = 0.203$) was the only variable included as the significant predictor of

**Table 2. Comparison of partial correlation coefficients between dementia mortality rate and each variable when the other three variables are controlled for.**

| Variables | Household size, aging, GDP PPP and urbanization were alternated as the predicting variable for calculating its relationship with dementia mortality rate while the other three independent variables were kept statistically constant. All data logarithmically transformed. | | | | | | | | | | | |
|---|---|---|---|---|---|---|---|---|---|---|---|---|
|  | Dementia Mortality | | | Dementia Mortality | | | Dementia Mortality | | | Dementia Mortality | | |
|  | R | P | df | r | p | df | R | p | df | r | p | df |
| Household size | -0.322** | <0.01 | 159 | - | - | - | - | - | - | - | - | - |
| Ageing | - | - | - | 0.154 | 0.051 | 159 | - | - | - | - | - | - |
| GDP PPP | - | - | - | - | - | - | -0.020 | 0.798 | 159 | - | - | - |
| Urbanization | - | - | - | - | - | - | - | - | - | 0.018 | 0.821 | 159 |

Significance level

** p< 0.01.

Data sources & definitions: Household size (the United Nations): the number of persons who make common provision of food, shelter and other essentials for living. Dementia Mortality Rate (World Health Organization): Calculated from Global Health Estimates. Ageing (the United Nations), measured with the Life Expectancy at 60 years old. Per capita GDP PPP (the World Bank): the per capita purchasing power parity (PPP) value of all final goods and services produced within a country in a given year. Urbanization (the World Bank): the percentage of population living in urban area.

**Table 3. Multiple linear regression showing predicting effects of independent variables and identify the significant predictors of dementia mortality.**

**Enter**

| | Dementia Mortality | | | |
|---|---|---|---|---|
| | Household size excluded, adjusted $R^2$ = 0.196 | | Household size included, adjusted $R^2$ = 0.282 | |
| Variable | Beta | Sig. | Beta | Sig. |
| Household Size | - | - | -0.357 | 0.001 |
| Ageing | 0.400 | 0.001 | 0.236 | 0.036 |
| GDP PPP | 0.043 | 0.735 | 0.063 | 0.624 |
| Urbanization | 0.039 | 0.704 | -0.076 | 0.474 |

**Stepwise**

| | Dementia Mortality | | | |
|---|---|---|---|---|
| | Household size excluded | | Household size included | |
| Model | Variable | Adjusted $R^2$ | Variable | Adjusted $R^2$ |
| 1 | Ageing | 0.203 | Household Size | 0.263 |
| 2 | - | - | Ageing | 0.289 |
| 3 | - | - | - | - |
| 4 | - | - | - | - |
| 5 | - | - | - | - |

Significance level: * p<0.05

** p< 0.01

***p< 0.001.

Data sources & definitions: Household size (the United Nations): the number of persons who make common provision of food, shelter and other essentials for living. Dementia Mortality Rate (World Health Organization): Calculated from Global Health Estimates. Ageing (the United Nations) measured with the Life Expectancy at 60 years old. Per capita GDP PPP (the World Bank): the per capita purchasing power parity (PPP) value of all final goods and services produced within a country in a given year. Urbanization (the World Bank): the percentage of population living in urban area.

DMR. However, when household size was included as an independent variable, household was selected as the variable having the greatest influence on DMR with $R^2$ = 0.263, while ageing was placed second increasing $R^2$ to 0.289. The other variables (GDP PPP and urbanization) were removed by the analysis as having no statistically significant influence on DMR.

Table 4 presents that, in general, household size is negatively associated with DMR in different country groupings. The highlight of these relationships was that household size was constantly in negative correlation to DMR. Small sample sizes in some categories of countries make differences between correlation coefficients found for them statistically insignificant and thus it is difficult to discern differences among those categories in the strength of correlations. However, in all categories, correlations are negative and often significantly different from zero. In economically developed country groupings, such as in the World Bank High income economics, WHO European Region and OECD all correlations are significant and moderately strong. Lower values of correlation coefficients in some groupings, especially those of lower income categories in the World Bank income classifications, may be a result of less variability in variables correlated when total ranges of their variation are artificially curtailed by categorisation. Averages of household size and dementia mortality in variously categorised countries show a tendency of larger household size to be associated with smaller dementia mortality.

**Table 4. Correlation of dementia mortality to household size in different country groupings.** Partial correlations are calculated when GDP, Urbanisation and Ageing are kept statistically constant.

| | Correlation coefficients | | | | Means (standard dev.) | |
|---|---|---|---|---|---|---|
| | **Pearson** | **Non-parametric** | **Partial** | **n** | **Household size** | **Dementia mortality** |
| **Country groupings** | | | | | | |
| Worldwide | -0.524*** | -0.579*** | -0.322** | 169 | 4.08(1.42) | 22.95(25.00) |
| World Bank income classifications | | | | | | |
| Low income | -0.287 | -0.274 | -0.159 | 32 | 5.32(1.11) | 8.40(4.15) |
| Low middle income | -0.271 | -0.221 | 0.008 | 44 | 4.68(1.06) | 13.47(10.12) |
| Upper middle income | -0.091 | -0.127 | -0.071 | 48 | 3.92(0.99) | 21.20(16.68) |
| High income | -0.623*** | -0.651*** | -0.460*** | 48 | 2.96(1.20) | 42.48(35.58) |
| WHO Regions | | | | | | |
| African Region (AFRO) | -0.351** | -0.437** | -0.192 | 42 | 5.01(1.19) | 9.54(6.18) |
| Eastern Mediterranean Region (EMRO) | -0.414 | -0.681** | 0.027 | 17 | 5.54(1.27) | 17.22(14.69) |
| European Region (EURO) | -0.519*** | -0.548*** | -0.101 | 52 | 3.02(1.04) | 40.94(34.71) |
| Pan-American Region (PARO) | -0.436** | -0.340 | -0.386 | 29 | 3.61(0.64) | 18.74(18.77) |
| South-East Asia Region (SEARO) | -0.589* | -0.823*** | -0.831* | 12 | 4.26(0.85) | 21.67(14.11) |
| Western Pacific Region (WPRO) | -0.300 | -0.472 | -0.406 | 19 | 4.28(1.16) | 19.39(16.41) |
| Countries grouped based on various factors | | | | | | |
| Asia Cooperation Dialogue (ACD) | -0.322 | -0.600*** | -0.338 | 30 | 4.68(1.38) | 17.33(14.20) |
| Asia-Pacific Economic Coop. (APEC) | -0.511* | -0.663** | -0.633* | 18 | 3.39(0.90) | 31.13(23.27) |
| Arab World | -0.372 | -0.544* | -0.018 | 18 | 5.63(1.09) | 17.04(14.47) |
| Latin America and the Caribbean (LAC) | -0.206 | -0.182 | -0.258 | 27 | 3.70(0.56) | 15.09(12.15) |
| Southern African Development Community (SADC) | -0.563* | -0.652* | -0.482 | 14 | 4.39(0.70) | 10.32(8.00) |
| Organisation for Economic Co-operation and Development (OECD) | -0.447** | -0.497** | -0.098 | 35 | 2.54(0.45) | 54.98(36.09) |

Significance level

* p<0.05

** p< 0.01

***p< 0.001.

Data sources & definitions: Household size (the United Nations): the number of persons who make common provision of food, shelter and other essentials for living. Dementia Mortality Rate (World Health Organization): Calculated from Global Health Estimates. Ageing (the United Nations) measured with the Life Expectancy at 60 years old. Per capita GDP PPP (the World Bank): the per capita purchasing power parity (PPP) value of all final goods and services produced within a country in a given year. Urbanization (the World Bank): the percentage of population living in urban area.

## 4. Discussion

The worldwide trend of increased DMR may have multiple aetiologies, which may act through multiple mechanisms. This study not only suggested that household size may be a major factor for dementia mortality at the population level, but also showed that household size was a determining risk factor overriding risk factors such as ageing, SES and urbanization. This study also revealed that the predicting effect of household size on dementia mortality was independent of the effects of other common risk factors, such as ageing, socio-economic status and urbanization.

Household relations are usually invested with the powerful norms, histories, and emotions that characterize family relationships [24]. From a life course perspective, they are related to the dementia risks in the essential pathways which are grounded on biological, psychological and social contributions [65]. In the industrialized societies, especially in the developed countries, most people grew up in core families with very limited number of playmates and little interaction with neighbours. This is different from how humans had adapted for flourishing

through early cooperative breeding [16, 17], and then evolved alloparental care [18], both of which laid the biological foundations of human love which may be heritable generation by generation [19]. The mismatch between the way we live now and how our ancestors did has been postulated as the risk factor for mental disorders in young generation [21, 22].

The lack of cure for dementia, and of accurate diagnoses of specific causal factors, has made it difficult to target preventative interventions [66]. It is well established that appropriate psychological, social support and physical care have been the key strategies for the healthcare for dementia patients [67]. Studies have revealed that, psychologically, family life increases levels of purpose in life [68, 69], which may reduce by 30% risk of dementia [70, 71]. Large household may offer the residents more social engagement in life which is protective against dementia [72, 73].

Compiled 2,200 years ago, *Huangdi Neijing* has been the fundamental doctrinal source of Asian medicine. It illustrates how emotions are associated with the visceral organs which are in charge of five qi's (translation: gas; meaning: emotions): happiness, anger, sadness, worry and fear. Among these five qi (emotions), only happiness makes the gas smooth [74], which keeps people healthy. In Western medicine, hundreds of years' exploration of the manifestation of emotions through physiological responses (mind—body interaction) [75–77] has suggested that the formation of mental experiences (emotions) is closely associated with bodily responses [78, 79].

Alzheimer's disease is the most common cause of dementia. The most prevalent hypothesis about this pathology is that it occurs when the beta-amyloid (toxic protein) begins to clump around neurons in the brain. The neurons degenerate, which leads to a decrease in synaptic plasticity and ultimately to cognitive decline. Oxytocin has been considered chemical responsible for targeting the removal or reduction of beta-amyloid or improvement of the cognitive ability of patients. An animal model study showed that toxic beta-amyloid damaged synaptic plasticity in mice's brains, but this beta-amyloid-induced impairment of hippocampal synaptic plasticity in mice was reversed by oxytocin in mice [80]. Interestingly, this study also revealed that oxytocin did not improve brain's synaptic plasticity in mice's brains if oxytocin was the only treatment [80]. This may imply that oxytocin was produced as an auto-immune response to the neuron blockage by toxic beta-amyloid in mice's brains. Similarly, another animal model study found that oxytocin strengthened social memory [81] and improved spatial memory [82] when mice were in their motherhood. In human studies, it has been reported that oxytocin selectively strengthened participants' memory for social stimuli depending on the participants' social contexts and individual attachment styles [82, 83].

Oxytocin has been implicated in many aspects of social functioning. The therapeutic effects of oxytocin have not only been explored in dementia prevention and treatment, but also have targeted the treatment of diseases for other aberrant social behaviour related disorders [84, 85], such as autism spectrum disorder [86–89], posttraumatic stress disorder [90, 91], schizophrenia [92, 93], and anxiety disorders [94, 95].

The nature of vascular dementia (VaD) is strongly associated with stroke [96, 97]. To date, there are few therapeutic options to protect cognitive decline arising from cerebrovascular diseases [98, 99]. Worldwide, prevention of strokes and management of post-stroke symptoms have been considered approaches to reduce the vascular dementia initiation [100]. Based on a number of studies in human and animal models, a systematic review conducted by Gutkowska and colleagues concluded that oxytocin has multiple roles in protecting cardiovascular system [101], which prevents VaD onset and might be the candidate treatment for VaD [99].

Frontotemporal dementia (FTD) is an umbrella term for a group of uncommon neurological diseases due to progressive damage to the frontal and/or temporal lobes of the brain. Empathy loss is one of hallmark symptoms of FTD [85, 88]. Oxytocin is an important mediator of

social behaviour, potentially enhancing empathy and prosocial behaviours [89]. Finger and colleagues reported that intranasal oxytocin improved the behavioural symptoms in FTD, including levels of apathy and expressions of empathy [89]. And the benefit or efficacy was associated with dosage and time related [89, 91]. This beneficial effect of oxytocin also improved emotional expression processing [93, 95], empathy [102], anxiety [103] and cooperative behaviour [104] in healthy adults and autism patients. Therefore, the mechanism of upregulated oxytocin mediation of empathy and behavioural deficits have been postulated as a potential treatment approach in FTD [89].

Large household promotes more interpersonal interactions between household members that may offer mind-body interaction which offers biological protection against dementia through the therapeutic effects of oxytocin [105–109]. At the same time, positive psychosocial well-being produced by large household may exert a beneficial slow-down on dementia development [70]. Family members who receive more family support may feel comprehensive positive psychological well-being [70, 110–112]. However, a greater meaning (purpose) of life may be the most important psychological resource to lower dementia risk [70, 110]. Sutin and colleagues explored the protective effects of psychological functioning (life satisfaction, optimism, mastery, purpose in life, positive affect) on preventing high risk population from developing dementia [70]. The results revealed that people from large household showed more psychological well-being leading to lower risk of dementia onset [70]. Interestingly, purpose in life explained 30% dementia risk and this protecting role was independent of the competing effects of other multiple risk factors, such as chronic disease and low physical activities, genetic risk, psychological distress and socioeconomic status [70].

Household residents may interact with each other more often to create life satisfaction [25, 113]. They may also share healthcare knowledge, encourage each other to establish healthy lifestyle and utilize health care services in an effective way [114–116]. People with positive psychological well-being tend to practise healthy lifestyle, have more knowledge of health risk factors and attend regular physical examination [27]. The protecting role of such positive psychological well-being has been postulated to decrease the risk in the development of breast cancer and general cancers [27, 36, 117, 118].

Large household is even more important for pre-dementia patient, especially the young onset dementia. Generally, young onset dementia present non-specific signs and symptoms at the early onset in young patient, but they are progressing and irreversible [119]. With household residents' observations and encouragement, the atypical dementia symptoms can be noticed by co-residential members, and accordingly they can have the examination in time, and proper treatment subsequently.

Additionally, from the perspective of evolution, a population with large household offers more chance to survive the natural selection. This produces the opportunity to have portion of a population with less fitness, for instance, dementia, removed through greater mortality rate without disturbing population's essential activities [33, 36, 37, 120–123]. In other words, the genetic background of dementia in such population may be more often eliminated from the population with large household without affecting population as a whole. Therefore, population with large household may have less genes/mutations of dementia to incur high mortality rate of dementia. Furthermore, a population with large household means less birth control and high total fertility rate which allows more biological variation in fertility [124]. A portion of this additional variation, however small, provides the opportunity for the natural selection [124].

Interestingly, large household has shown its consistent and significant, but inverse correlations to dementia mortality rates in the developed world. Several unique phenomena in the developed world can explain this interesting relationship: 1) The fertility rate keeps falling

down leading to small household/family size. 2) The cultural doctrines, especially individualism and independence, have reduced the interactions between people living in the developed world. 3) Importantly, dementia care delivery in the developed world is primarily through individual home support service, because dementia patients live with small family/household, or through nursing homes [125]. While in the developing world people live with big family/household for most of their life and they would receive the "informal" healthcare from family members and/or household residents [126], instead of the "formal" care through nursing homes. The fact that people live in bigger households in developing countries is a combined result of the economic situation and social customs. With lower incomes and less availability of housing people are more likely to share accommodation while social customs demand respect and care for elderly people, who, in the situation of low incomes cannot afford to have living arrangements independent of members of younger generations. In developed countries higher incomes, greater availability of accommodation, higher mobility and individualistic lifestyles of adults, together with well-resourced retirement situations of older persons allow individuals or couples to have separate households. In this situation, social interactions with other individuals become sporadic personal choices rather than a daily necessity.

## 5. Strengths and limitations in this study

Due to the inherent uncertainties of the worldwide estimates provided by the WHO and the World Bank, our results must be seen as showing relations between published estimates, not necessarily precisely measured actual phenomena. The relations, however, are the closest available approximation of actual situation.

Little work has been done on the dementia epidemiology studies, which may be due to extremely low onset rate making data collection difficult. For example, worldwide, mortality rate of dementia is only 26.61 per 100,000, and this presence is not noticeable. Therefore, the low prevalence rate of dementia would require unaffordable sample size for identifying small household as the potential risk factor for dementia in individual based epidemiological or laboratory approaches. Ecological studies are based on aggregated quantitative data zooming in the rare presence of dementia 100,000 times, which makes dementia presence noticeable for analysing the potential effects of dementia risk-modifying factors at population level. This also suggests the necessity of engaging ecological study into the epidemiology studies of rare chronic diseases such as dementia and cancer [28, 29, 36] and Type 1 diabetes [33].

Due to the nature of the cross-sectional data, a couple of intrinsic limitations should be mentioned. Firstly, the results in this study only showed the relationship between household size and dementia mortality rate as correlational, instead of causal. Secondly, the results based on the ecological approach in this work are subject to the "ecological fallacy". Therefore, the protective role of large household size may not always hold true for each individual to predict their dementia specific death risk. Thirdly, dementia mortality has been associated with multiple risk factors which could have confounded the relationship between large household/family size and dementia mortality rate in this study. These factors include, but are not limited to, hypertension, obesity, head injury and hearing loss in middle life, and smoking, depression, diabetes and social isolation in later life. However, these factors have not been well established for explaining dementia mortality. Also considering unavailability of the data on these factors at population level, we cannot include them as the potential confounders for ruling out their competition with the detrimental effect of small household/family size on dementia mortality in this study. Finally, the data employed in this study might be crude. The WHO, United Nations and World Bank may have made some random errors arising from the methodologies used for collecting and aggregating the data.

Regardless of the strength and limitations of the data quality, we have showed that countries with large household size had lower dementia mortality rate in different data analysis models. The findings in this study may have shed light for further research into the subject with exposure based longitudinal cohort studies at population level. Accordingly, this may lead to far reaching public health implications in dementia and its prevention.

## 6. Conclusions

Independent of ageing, urban lifestyle and low socio-economic status, large household has been shown to have significant protective role against dementia mortality in this study. This forward-thinking study approach will help shed light on the further study into the protective role of positive psychological well-being against dementia. As part of dementia prevention, when large household/family size is impossible to achieve, healthcare practitioners should encourage people to increase their positive interactions with people from their neighbourhood or other fields.

## Supporting information

**S1 Data. S1 A whole set of data for this study.**
(XLSX)

## Acknowledgments

The authors express appreciation to Dr Cherian Varghese from the Management of on communicable Diseases, Disability, Violence and Injury Prevention Department (NVI) of World Health Organization for the assistance in locating the data.

The authors also appreciate the great assistance from the reviewers in improving this study.

## Author Contributions

**Conceptualization:** Wenpeng You, Maciej Henneberg.

**Data curation:** Wenpeng You, Maciej Henneberg.

**Formal analysis:** Wenpeng You, Maciej Henneberg.

**Investigation:** Wenpeng You, Maciej Henneberg.

**Methodology:** Wenpeng You, Maciej Henneberg.

**Project administration:** Wenpeng You, Maciej Henneberg.

**Resources:** Wenpeng You, Maciej Henneberg.

**Software:** Wenpeng You, Maciej Henneberg.

**Supervision:** Maciej Henneberg.

**Validation:** Wenpeng You, Maciej Henneberg.

**Visualization:** Wenpeng You, Maciej Henneberg.

**Writing – original draft:** Wenpeng You, Maciej Henneberg.

**Writing – review & editing:** Wenpeng You, Maciej Henneberg.

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
