## [Decision Letter · Decision Letter 0]

17 Sep 2021

PONE-D-21-26012Large household reduces dementia mortality: indications for patient carePLOS ONE

Dear Dr. Wenpeng You,

Thank you for submitting your manuscript to PLOS ONE. After careful consideration, we feel that it has merit but does not fully meet PLOS ONE’s publication criteria as it currently stands. Therefore, we invite you to submit a revised version of the manuscript that addresses the points raised during the review process.

We look forward to receiving your revised manuscript.

Kind regards,

Wen-Wei Sung, M.D., Ph.D.

Academic Editor

PLOS ONE

- https://file.scirp.org/Html/1-2470163_81611.htm

The text that needs to be addressed involves the Results section.

In your revision ensure you cite all your sources (including your own works), and quote or rephrase any duplicated text outside the methods section. Further consideration is dependent on these concerns being addressed.

Reviewers' comments:

Reviewer's Responses to Questions

**Comments to the Author**

1. Is the manuscript technically sound, and do the data support the conclusions?

Reviewer #1: Yes

Reviewer #2: Yes

Reviewer #3: Yes

2. Has the statistical analysis been performed appropriately and rigorously? 

Reviewer #1: Yes

Reviewer #2: Yes

Reviewer #3: Yes

3. Have the authors made all data underlying the findings in their manuscript fully available?

Reviewer #1: No

Reviewer #2: Yes

Reviewer #3: No

4. Is the manuscript presented in an intelligible fashion and written in standard English?

Reviewer #1: Yes

Reviewer #2: Yes

Reviewer #3: Yes

5. Review Comments to the Author

Reviewer #1: This a fascinating and worthwhile article.

Databases

Please report the assessment of the data compilers of the original documents and any technical reports on how the data on household size was collected and the reliability and validity of the measurement of household size.

Similarly please comment in detail how the diagnosis of dementia death was made in the various countries you reported and any reliability and validity data. Was it based on samples and if so what was the sample size?

What proportion of individuals and what are the numbers of individuals who were diagnosed with dementia in the countries you assessed? Some less developed countries have fewer people in the relevant age groups so the confidence intervals if based on samples would be wider.

Please comment in detail on Table 4 and differences between countries for which you have an explanation.

My comments are directed to strengthening your paper.

Reviewer #2: Well done!

The thematic is very important and we know that is a lack of evidence on this particular subject related to Large households/families to prevent dementia.

I have some considerations to the article that should be clarified

Abstract: Please mention the type of study design such as cross-sectional/correlation design. The method section should not have any of your results as in your last paragraph of method section.

SES: This abbreviation did not mention before.

Introduction:

1- It is easy to read, written in a professional way, and has plenty of information.

2- Many unnecessary conjunctions between paragraphs.

3- typos before reference (20, 21) in in small communities.

4- Please can you add a paragraph regarding country grouping (low, middle, and high-income country). do you have statistics on the dementia rate?

Methodology:

1- Study design?

2- The data source is not clear, need to clarify how it was done?

3- The sample size is not well described - Power analyses?

Results:

1- SMR variables???

2- Table 3. 0.000 replace it with 0.001, please add R change.

3- All abbreviations in all Tables should be mention as a footnote. Mainly Table 4

Discussion:

1-The therapeutic effects of Oxytocin. Why you added. Is your study investigate the level of Oxytocin? You discussed previous studies regarding Oxytocin and its effect on dementia (VaD, FTD...) If not I suggest removing this or a better explain why you mention that in the first paragraph.

2- You mention this statement: Similarly, five (5) studies conducted by Lambert and co-workers also identified the

independent relationship between the meaning of life and family support among your people [116]. I returned to this paper and I read it carefully, Unfortunately, it is about the use of social media in healthcare settings not as you mention.

3- Can you add the year after you mention the authors such as Stutin and colleagues (??).

4- Finally, the last paragraph regarding developed countries before the limitation section. I like the explanation, but to add more strength can you add more explanation between developing countries (Low- and middle-income) and developed countries.

5- Good luck.

Reviewer #3: 1) Please consider reversing the order of x-axis values so that it shows a negative association between household size and dementia mortality rates as reported in text.

2) Please provide a table for the descriptive statistics of the variables used for the study: basic measures of central tendency and variability for the whole countries and by different country categorizations.

3) The correlation between household size and DMR reported on page 7 is -.6034, but the correlation for “worldwide” reported in Table 4 is -.524. Please clarify the difference between these two numbers.

4) The main research question is to assess whether large household has the inhibitory role in lowering the risk for the residents “to develop dementia”. However, the dependent variable of the study is DMR, which is different from “development of dementia”. Please clarify why DMR, not some other measures more closely measuring the development of dementia. Perhaps, Alzheimer's disease morbidity can be explored together to draw a bigger picture but not sure about data availability.

5) The top part of Table 2 and the bottom part of Table 2 provide the same numbers. And the numbers on the first line of page 8 do not exactly match the numbers in Table 2. FYI, ageing is not significant at the .05 level of significance since the p-value reported in Table 2 is .051, which is >.050. Please double-check.

6) Table 4 reports valuable information about the variation in the association between household size and DMR. But it seems that the negative association between household size and DMR is only observed among high income countries (-.623, n=43), which is contradictory to the main finding of the study (Table 3). The results for countries grouped based on various factors somewhat agree with this result. Please discuss potential sources of the discrepancy.

7) Also wondering if the negative association between household size and DMR controlling for ageing, GDP, and urbanization, is still observed for those countries with high income. It would be very informative if the regression results with all variables (household size, ageing, GDP, and urbanization) can be reported for each country group, especially based on World Bank income classification, as well.

8) It is unclear why using three different categorizations for countries. Other than World Bank one, it seems that the other categorizations provide more heterogenous grouping than homogenous.

6. PLOS authors have the option to publish the peer review history of their article (what does this mean?). If published, this will include your full peer review and any attached files.

Reviewer #1: **Yes: **Roger E. Thomas

Reviewer #2: No

Reviewer #3: No

---

## [Author Response · Author response to Decision Letter 0]

7 Oct 2021

Please see the uploaded document, titled 1. PLOS ONE 1 to 1 Response.

---

## [Decision Letter · Decision Letter 1]

25 Oct 2021

PONE-D-21-26012R1Large household reduces dementia mortality: indications for patient carePLOS ONE

Dear Dr. Wenpeng You,

Thank you for submitting your manuscript to PLOS ONE. After careful consideration, we feel that it has merit but does not fully meet PLOS ONE’s publication criteria as it currently stands. Therefore, we invite you to submit a revised version of the manuscript that addresses the points raised during the review process.

We look forward to receiving your revised manuscript.

Kind regards,

Wen-Wei Sung, M.D., Ph.D.

Academic Editor

PLOS ONE

Reviewers' comments:

Reviewer's Responses to Questions

**Comments to the Author**

1. If the authors have adequately addressed your comments raised in a previous round of review and you feel that this manuscript is now acceptable for publication, you may indicate that here to bypass the “Comments to the Author” section, enter your conflict of interest statement in the “Confidential to Editor” section, and submit your "Accept" recommendation.

Reviewer #1: (No Response)

Reviewer #2: All comments have been addressed

Reviewer #3: All comments have been addressed

2. Is the manuscript technically sound, and do the data support the conclusions?

Reviewer #1: No

Reviewer #2: Yes

Reviewer #3: Yes

3. Has the statistical analysis been performed appropriately and rigorously? 

Reviewer #1: Yes

Reviewer #2: Yes

Reviewer #3: Yes

4. Have the authors made all data underlying the findings in their manuscript fully available?

Reviewer #1: No

Reviewer #2: Yes

Reviewer #3: Yes

5. Is the manuscript presented in an intelligible fashion and written in standard English?

Reviewer #1: Yes

Reviewer #2: Yes

Reviewer #3: Yes

6. Review Comments to the Author

Reviewer #1: As this is a data base study the vaildity and reliability of the data are key The authors have not provided any new data on the reliability and validity of their data bases

Page 2 They provide a reference to the WHO mortality rate, WHO Technical Paper [ref 43) but do not provide key data from that report.

Page 5 Please comment on data validity and reliability and missing data

Page 6 You refer to the World Bank published data on GDP and urbanisation but do not explain in detail the potential confounders you mention

Page 6. "certain criteria for completeness and quality" Please provide details and how this affects your study

Page 12 You wrote "Complied 2,200 years ago Huangdi Neijing" ... What are the data for the reliability and validity of the following statements?

"Large household promotes more mind-body interaction which offers biological protection" What is the evidence for this unsupported unsubstantiated statement.

Pleased read your manuscript carefully and remove any unsubstantiated statements, especially to be found in the Discussion section.

Reviewer #2: All my comments have been answered prperly. I would like to take this chance to congrat the authors for their valuable effort. Great job. Best of my luck.

Reviewer #3: Thank you for considering my comments to revise the manuscript. The manuscript has substantially improved.

7. PLOS authors have the option to publish the peer review history of their article (what does this mean?). If published, this will include your full peer review and any attached files.

Reviewer #1: No

Reviewer #2: No

Reviewer #3: No

---

## [Author Response · Author response to Decision Letter 1]

15 Nov 2021

Please see detailed response in the attached document titled, 1 to 1 Response to Reviewer 1 comments.

---

## [Decision Letter · Decision Letter 2]

6 Dec 2021

PONE-D-21-26012R2Large household reduces dementia mortality.PLOS ONE

Dear Dr. Wenpeng You,

Thank you for submitting your manuscript to PLOS ONE. After careful consideration, we feel that it has merit but does not fully meet PLOS ONE’s publication criteria as it currently stands. Therefore, we invite you to submit a revised version of the manuscript that addresses the points raised during the review process.

We look forward to receiving your revised manuscript.

Kind regards,

Wen-Wei Sung, M.D., Ph.D.

Academic Editor

PLOS ONE

Journal Requirements:

Additional Editor Comments:

According to the review report of reviewer #1, previous revision did not meet the criteria for publication. Therefore, I added more reviewers for evaluation and the decision was made base on total of five reviewers. In this major revision, please carefully answer all questions from reviwer #1 and #6 and submit revised MS for further consideration. 

Reviewers' comments:

Reviewer's Responses to Questions

**Comments to the Author**

1. If the authors have adequately addressed your comments raised in a previous round of review and you feel that this manuscript is now acceptable for publication, you may indicate that here to bypass the “Comments to the Author” section, enter your conflict of interest statement in the “Confidential to Editor” section, and submit your "Accept" recommendation.

Reviewer #1: (No Response)

Reviewer #4: All comments have been addressed

Reviewer #5: (No Response)

Reviewer #6: (No Response)

Reviewer #7: All comments have been addressed

2. Is the manuscript technically sound, and do the data support the conclusions?

Reviewer #1: No

Reviewer #4: Yes

Reviewer #5: Yes

Reviewer #6: Yes

Reviewer #7: (No Response)

3. Has the statistical analysis been performed appropriately and rigorously? 

Reviewer #1: No

Reviewer #4: Yes

Reviewer #5: Yes

Reviewer #6: Yes

Reviewer #7: (No Response)

4. Have the authors made all data underlying the findings in their manuscript fully available?

Reviewer #1: No

Reviewer #4: Yes

Reviewer #5: Yes

Reviewer #6: No

Reviewer #7: (No Response)

5. Is the manuscript presented in an intelligible fashion and written in standard English?

Reviewer #1: Yes

Reviewer #4: Yes

Reviewer #5: Yes

Reviewer #6: Yes

Reviewer #7: (No Response)

6. Review Comments to the Author

Reviewer #1: Thank you for your updated version.

As I mentioned with your first submission the assessment of the risk of bias of the databases is crucial to your study. There are minimal changes in your manuscript. You have not assessed the risk of bias in your databases, which would have required identifying key documents about the methods they used and their assessments of reliability from the database constructors.

You have not considered all the possible confounders.

You do not assess the implications of your data. For example you define aging as life expectancy at 60 and found the median is 19 years and the SD = 3.1 years. This implies a median age at death of 79, which would imply your population is heavily influenced by wealthier countries. The SD at 3.1 is small so can you draw conclusions based on this small SD?

Similarly you found median household size was 3.9 with an SD = 1.4 Can you draw conclusions with such a small SD?

I suggested that you remove undocumented and unsubstantiated statements but they remain.

You have found an important idea but have not proven your hypothesis with data of sufficient low risk of bias.

Reviewer #4: The author responses well for the rebuttal and makes a great change for the revised manuscript. I have no further comment or suggestion for this manuscript.

Reviewer #5: This ecological study showed that household size was an independent factor associated with dementia mortality at the population level, which provided evidence that social interaction may improve the prognosis of dementia.

The authors do their best to reduce any possible biases. I have no issue to highlight.

Reviewer #6: The manuscript is well-written; however, I would like to clarify some of the points regarding the study:

Major issues:

1, In the introduction, the authors mentioned, “…., The above considerations directed us to try to identify possible contributing factors for dementia from the evolutionary perspective.” However, the current study seemed to focus on “whether smaller household could serve as a risk factor for people dying of dementia”, instead of “whether smaller household could serve as a risk factor for dementia”. Though the authors had avoided using the sentence “developing dementia” in the revised version, the quoted sentence might be confusing for readers, for the two aforementioned concepts are not exactly same.

I would recommend the authors provide a clearer explanation in the manuscript regarding the rationale of utilizing DMR manifesting the development of dementia.

2, Frailty bias could exist and should be addressed:

Severe adverse statuses or comorbidities, such as cancer or organ transplantation could massively affect mortality, influencing risk for death occurrence. Could you please comment why the issue was not considered as one of the variables in the current study?

Minor issues:

1, For reference 43, according to the formal name of the document, I think the correct title of this technical paper should be “WHO methods and data sources for country-level causes of death 2000-2016”.

(According to the WHO technical paper, retrieved from: https://www.who.int/healthinfo/global_burden_disease/GlobalCOD_method_2000-2016.pdf)

2, For information related to dementia mortality, the dataset utilized by the authors was based on Global Health Estimates 2016; however, currently the GHE report has been updated to 2019 version. Hence it might be difficult for the readers to directly find the dataset you have retrieved. Therefore, I would recommend the authors provide the exact website address to make readers more easily to access the key data identifying the number of dementia death and other information.

Reviewer #7: (No Response)

7. PLOS authors have the option to publish the peer review history of their article (what does this mean?). If published, this will include your full peer review and any attached files.

Reviewer #1: No

Reviewer #4: No

Reviewer #5: No

Reviewer #6: No

Reviewer #7: No

---

## [Author Response · Author response to Decision Letter 2]

11 Dec 2021

Please refer to the document, titled 1 to 1 response to review comments

---

## [Decision Letter · Decision Letter 3]

26 Dec 2021

PONE-D-21-26012R3Large household reduces dementia mortality: A cross-sectional data analysis of 183 populationsPLOS ONE

Dear Dr. Wenpeng You,

Thank you for submitting your manuscript to PLOS ONE. After careful consideration, we feel that it has merit but does not fully meet PLOS ONE’s publication criteria as it currently stands. Therefore, we invite you to submit a revised version of the manuscript that addresses the points raised during the review process.

There are still some concerns need further revision. Please revise this manuscript accordingly.

We look forward to receiving your revised manuscript.

Kind regards,

Wen-Wei Sung, M.D., Ph.D.

Academic Editor

PLOS ONE

Reviewers' comments:

Reviewer's Responses to Questions

**Comments to the Author**

1. If the authors have adequately addressed your comments raised in a previous round of review and you feel that this manuscript is now acceptable for publication, you may indicate that here to bypass the “Comments to the Author” section, enter your conflict of interest statement in the “Confidential to Editor” section, and submit your "Accept" recommendation.

Reviewer #1: (No Response)

Reviewer #6: (No Response)

2. Is the manuscript technically sound, and do the data support the conclusions?

Reviewer #1: No

Reviewer #6: Yes

3. Has the statistical analysis been performed appropriately and rigorously? 

Reviewer #1: No

Reviewer #6: Yes

4. Have the authors made all data underlying the findings in their manuscript fully available?

Reviewer #1: No

Reviewer #6: No

5. Is the manuscript presented in an intelligible fashion and written in standard English?

Reviewer #1: Yes

Reviewer #6: Yes

6. Review Comments to the Author

Reviewer #1: The authors have made minimal changes. This is a retrospective databases study. Therefore, assertions of causal relationships cannot be made.

1. The authors persist is making predictive statements:

Abstract:

"Large households/families create more positive psychological well-being"

"Large household was an independent predictor."

Text

"large household protects against dementia mortality."

large household "most significant predictor of DMR."

2. The section on oxytocin.

In this section the authors assert that oxytocin is the mediator molecule between their assertions and refer to "a stream of studies..." Could the authors please provide detailed analyses of studies with serum levels and differences in serum levels of oxytocin with different household sizes and rates of interaction (which again would be correlational relationships and not causal) or delete this section on oxytocin. Neurobiologists would ask for better data than are presented here.

3. Data elements.

The authors used DMR, "ageing" GDP and urbanisation as their variables. Could the authors please list potential known confounders (and potential confounders unknown in their database and control variables) which could affect their results.

Reviewer #6: The authors have fully addressed all raised issues, and I have no further major concerns.

However, it should be noticed that in the updated reference 41 (namely, the dataset of GHE utilized by the author), the website is revamped and it seems the original dataset is no longer available.

Hence, I think this should be stated in the manuscript to remind the readers that the version 2016 is no longer publicly accessible.

7. PLOS authors have the option to publish the peer review history of their article (what does this mean?). If published, this will include your full peer review and any attached files.

Reviewer #1: No

Reviewer #6: No

---

## [Author Response · Author response to Decision Letter 3]

5 Jan 2022

Please refer to the attachment titled 1 to 1 response_ 02012022. 

This allows us to maintain the formatting of the response to help the editor and reviewers to read. However. we are pasting it below: 

PONE-D-21-26012R3

Large household reduces dementia mortality: A cross-sectional data analysis of 183 populations

PLOS ONE

6. Review Comments to the Author

Reviewer #1: The authors have made minimal changes. This is a retrospective databases study. Therefore, assertions of causal relationships cannot be made.

Authors: This had been listed as one of the study limitations (first). 

1. The authors persist is making predictive statements:

Abstract: Thanks a lot for this comment. 

The predictive statements have been toned down or deleted from the manuscript. 

"Large households/families create more positive psychological well-being"

Authors: We stated that “Large households/families create more positive psychological well-being—-" in the Background of the Abstract. Now it is amended as:

Large households/families may create more happiness and offer more comprehensive healthcare between the members. We analyse the protecting role against dementia mortality through examining the relationship between household/family size and dementia mortality rate at population level. 

This paragraph has been elaborated in the following paragraph copied from the manuscript: 

It is well researched that positive psychological well-being has been implicated in health across adulthood [1]. Household creates a social environment which is salient to maintain health for the co-residential members. On a daily basis, the individual members encounter this environment, play their social role and enjoy the social relations [2]. Moreover, studies also showed that large household offers the residents the subjective happiness [3] leading to low risk for residents to develop various cancers [4], for instance female breast cancer [5, 6] and ovarian cancer [7]. Subjective happiness was associated with mental health significantly stronger than with physical health in people with disabilities [8] and hospital patients [9]. A recent study revealed that greater household size has the protecting role against children developing mental health disorders [10]. 

 "Large household was an independent predictor."

Authors: This sentence was included in the Result of the Abstract below: 

Regardless of the contribution of ageing, socio-economic status and urban lifestyle to dementia mortality, large household was an independent predictor of dementia mortality (r = −0.331, p <0.001) in partial correlation analysis.

Thanks for this comment. We found that we did not state this properly. We meant to say, 

Small (NOT large) household was an independent predictor/risk factor for dementia mortality [independent of ageing, socio-economic status and urban lifestyle]. We apologize for this confusion. 

Now it has been rewritten and included in the Abstract in consideration of the context: 

When we controlled for the contribution of ageing, socio-economic status and urban lifestyle in partial correlation analysis, large household was still in inverse and significant correlation to dementia mortality (r = −0.331, p <0.001). This suggested that, statistically, large household protect against dementia mortality regardless of the contributing effects of ageing, socio-economic status and urban lifestyle. 

Text

"large household protects against dementia mortality."

Authors: This sentence was in the Conclusion section of the Abstract: 

Independent of ageing, urbanization and GDP, large household protects against dementia mortality. 

Now it has been toned down: 

While acknowledging that ageing, urban lifestyle and greater GDP are associated with dementia mortality, this study suggested that, at population level, small household size was another risk factor for dementia mortality. 

large household "most significant predictor of DMR." 

Authors: We stated that "most significant predictor of DMR." because of our data analysis results. In this study, standard multiple linear regression analysis was performed. This analysis model regressed multiple variables while simultaneously removing those that were not important/significant, but at the same time those most significant (important) predictors were maintained in order of their influencing effects. For instance, in Table 3 - Stepwise model, when household size was included as one of the predicting variables (four in total: household size, ageing, urban lifestyle and GDP), it became the most significant/important risk factor for dementia mortality rate. We described this in the Result: 

However, when household size was included as an independent variable, household was selected as the variable having the greatest influence on DMR with R2 = 0.263, while ageing was placed second increasing R2 to 0.289.

Before this, this analysis model was introduced in 2.4 Data analysis of the manuscript: 

3. Standard multiple linear regression (enter and stepwise) was performed to visualize the correlation between DMR and each predicting factor and identify the most significant predictor(s) of DMR respectively.

2. The section on oxytocin.

In this section the authors assert that oxytocin is the mediator molecule between their assertions and refer to "a stream of studies..." Could the authors please provide detailed analyses of studies with serum levels and differences in serum levels of oxytocin with different household sizes and rates of interaction (which again would be correlational relationships and not causal) or delete this section on oxytocin. Neurobiologists would ask for better data than are presented here.

Authors: The paragraph discussing the mediating effects of oxytocin has been removed. 

Much appreciated for providing this important comment from the perspective of neurobiologist. This makes this study reporting and discussion more rigorous. A stream of studies did introduce the therapeutic role of oxytocin on different diseases, but it seems that there is not enough substantiated scientific support, at least at the molecular level. The following two paragraphs have been deleted from the Discussion: 

Oxytocin is a hormone and a neurotransmitter that is associated with social bonding, such as empathy, trust, sexual activity, group bonding and relationship-building [11]. A stream of studies in the last decade reported that oxytocin release is not only associated with giving birth [12] and lactation [13], but also with daily interactions between non-kin household residents and/or family members, such as spouses [14-16], mother and children [17], father and children [18] and co-residential household members [19-21]. Oxytocin can keep family members and household residents happy and loyal to each other [22, 23], which may bring more positive psychological well-being to the family members. Regardless of cultural backgrounds [3], people from large household, especially from the same family have more life satisfaction [3, 24] which may lead to more oxytocin production within the hypothalamo-pituitary magnocellular systems. A self-reinforcing cycle is formed between more household interactions and more oxytocin production [11]. 

Oxytocin has been implicated in many aspects of social functioning. The therapeutic effects of oxytocin have not only been explored in dementia prevention and treatment, but also have targeted the treatment of diseases for other aberrant social behaviour related disorders [25], such as autism spectrum disorder [26, 27], posttraumatic stress disorder [28], schizophrenia [29], and anxiety disorders [30]. 

3. Data elements.

The authors used DMR, "ageing" GDP and urbanisation as their variables. Could the authors please list potential known confounders (and potential confounders unknown in their database and control variables) which could affect their results.

Authors: In statistics-based epidemiology studies, those best-established contributing variables are included in data analysis models as the existing detrimental or beneficial factors for the studied health challenge. These factors may compete the studied (independent) variable for predicting the specific health challenge (dependent variable). As the health effects of these factors have not been examined in the clinical trials, the relationship between the health challenge between each of the best-established variables is potential or correlational (not causal). During the study result reporting and discussing, those best- established variables are generally called potential (due to their effects not confirmed in clinical trials) confounders (due to their competition with the specific studied independent variable). 

In this study, our literature review showed that ageing, GDP and urban lifestyle are well established/known risk factors. However, the aim of this study is to advance small household/family size may be another risk factor for the increase of dementia mortality rate. In other words, small household/family size may exert some detrimental effects on dementia mortality except those contributed by ageing, GDP and urban lifestyle. We achieved this goal through data analyses in partial correlation when we statistically controlled for the three potential confounders (ageing, GDP and urban lifestyle), and in enter linear regression when we included all the four variables (household size, ageing, GDP and urban lifestyle). 

Some factors could be the risk factors as well, such as hypertension, obesity, head injuries and hearing loss in middle life, and smoking, depression, diabetes and social isolation in later life. However, due to data availability and/or low levels of establishment as risk factors for dementia mortality in previous studies. We could not include them as the potential confounders in our data analyses. This question has led us to list an additional study limitation: 

Thirdly, dementia mortality has been associated with multiple risk factors which could confound the relationship between large household/family size and dementia mortality rate in this study. These factors include, but not limited to, hypertension, obesity, head injury and hearing loss in middle life, and smoking, depression, diabetes and social isolation in later life. However, these factors have not been well established for explaining dementia mortality. Also considering unavailability of the data on these factors at population level, we cannot include them as the potential confounders for ruling out their competition with the detrimental effect of small household/family size on dementia mortality in this study. 

Sorry, we did not collect and keep the data of the variables which were not considered as the major risk factors for dementia mortality rate.

Reviewer #6: The authors have fully addressed all raised issues, and I have no further major concerns.

However, it should be noticed that in the updated reference 41 (namely, the dataset of GHE utilized by the author), the website is revamped and it seems the original dataset is no longer available.

Hence, I think this should be stated in the manuscript to remind the readers that the version 2016 is no longer publicly accessible.

Authors: Thanks for bringing this important comment again. 

With the assistance of Dr Cherian Varghese, the Coordinator, Management of Noncommunicable Diseases (MND), the hyperlink of the data source has been obtained: 

https://www.who.int/healthinfo/global_burden_disease/GHE2016_Deaths_WBInc_2000_2016.xls

It has been included in the References of the manuscript. 

References: 

1. Howell RT, Kern ML, Lyubomirsky S: Health benefits: Meta-analytically determining the impact of well-being on objective health outcomes. Health Psychology Review 2007, 1(1):83-136.

2. Hughes ME, Waite LJ: Health in household context: Living arrangements and health in late middle age. Journal of health and social behavior 2002, 43(1):1.

3. Nan H, Ni MY, Lee PH, Tam WW, Lam TH, Leung GM, McDowell I: Psychometric evaluation of the Chinese version of the Subjective Happiness Scale: evidence from the Hong Kong FAMILY Cohort. International journal of behavioral medicine 2014, 21(4):646-652.

4. You W, Rühli FJ, Henneberg RJ, Henneberg M: Greater family size is associated with less cancer risk: an ecological analysis of 178 countries. BMC cancer 2018, 18(1):924.

5. Bai A, Li H, Huang Y, Liu X, Gao Y, Wang P, Dai H, Song F, Hao X, Chen K: A survey of overall life satisfaction and its association with breast diseases in Chinese women. Cancer medicine 2016, 5(1):111-119.

6. You W, Symonds I, Rühli FJ, Henneberg M: Decreasing birth rate determining worldwide incidence and regional variation of female breast Cancer. Advances in Breast Cancer Research 2018, 7(01):1-14.

7. You W, Symonds I, Henneberg M: Low fertility may be a significant determinant of ovarian cancer worldwide: an ecological analysis of cross-sectional data from 182 countries. Journal of ovarian research 2018, 11(1):1-9.

8. Van Campen C, Iedema J: Are persons with physical disabilities who participate in society healthier and happier? Structural equation modelling of objective participation and subjective well-being. Quality of Life Research 2007, 16(4):635.

9. Mukuria C, Brazier J: Valuing the EQ-5D and the SF-6D health states using subjective well-being: a secondary analysis of patient data. Social Science & Medicine 2013, 77:97-105.

10. Grinde B, Tambs K: Effect of household size on mental problems in children: results from the Norwegian Mother and Child Cohort study. BMC psychology 2016, 4(1):31.

11. Magon N, Kalra S: The orgasmic history of oxytocin: Love, lust, and labor. Indian journal of endocrinology and metabolism 2011, 15(Suppl3):S156.

12. Takayanagi Y, Yoshida M, Bielsky IF, Ross HE, Kawamata M, Onaka T, Yanagisawa T, Kimura T, Matzuk MM, Young LJ: Pervasive social deficits, but normal parturition, in oxytocin receptor-deficient mice. Proceedings of the National Academy of Sciences of the United States of America 2005, 102(44):16096-16101.

13. White‐Traut R, Watanabe K, Pournajafi‐Nazarloo H, Schwertz D, Bell A, Carter CS: Detection of salivary oxytocin levels in lactating women. Developmental psychobiology 2009, 51(4):367-373.

14. Carmichael MS, Humbert R, Dixen J, Palmisano G, Greenleaf W, Davidson JM: Plasma oxytocin increases in the human sexual response. The Journal of Clinical Endocrinology & Metabolism 1987, 64(1):27-31.

15. Carmichael MS, Warburton VL, Dixen J, Davidson JM: Relationships among cardiovascular, muscular, and oxytocin responses during human sexual activity. Archives of sexual behavior 1994, 23(1):59-79.

16. Gordon Jr G, Burch RL, Platek SM: Does semen have antidepressant properties? Archives of Sexual Behavior 2002, 31(3):289-293.

17. Kendrick KM: The neurobiology of social bonds. Journal of neuroendocrinology 2004, 16(12):1007-1008.

18. Weisman O, Zagoory-Sharon O, Feldman R: Oxytocin administration to parent enhances infant physiological and behavioral readiness for social engagement. Biological psychiatry 2012, 72(12):982-989.

19. MacDonald K, MacDonald TM: The peptide that binds: a systematic review of oxytocin and its prosocial effects in humans. Harvard review of psychiatry 2010, 18(1):1-21.

20. Van IJzendoorn MH, Bakermans-Kranenburg MJ: A sniff of trust: meta-analysis of the effects of intranasal oxytocin administration on face recognition, trust to in-group, and trust to out-group. Psychoneuroendocrinology 2012, 37(3):438-443.

21. Wudarczyk OA, Earp BD, Guastella A, Savulescu J: Could intranasal oxytocin be used to enhance relationships? Research imperatives, clinical policy, and ethical considerations. Current opinion in psychiatry 2013, 26(5):474.

22. Insel TR, Hulihan TJ: A gender-specific mechanism for pair bonding: oxytocin and partner preference formation in monogamous voles. Behavioral neuroscience 1995, 109(4):782.

23. Young LJ, Murphy Young AZ, Hammock EA: Anatomy and neurochemistry of the pair bond. Journal of Comparative Neurology 2005, 493(1):51-57.

24. Angeles L: Children and life satisfaction. Journal of happiness Studies 2010, 11(4):523-538.

25. Fineberg SK, Ross DA: Oxytocin and the social brain. Biological psychiatry 2017, 81(3):e19.

26. Parker KJ, Oztan O, Libove RA, Sumiyoshi RD, Jackson LP, Karhson DS, Summers JE, Hinman KE, Motonaga KS, Phillips JM: Intranasal oxytocin treatment for social deficits and biomarkers of response in children with autism. Proceedings of the National Academy of Sciences 2017, 114(30):8119-8124.

27. Alvares GA, Quintana DS, Whitehouse AJ: Beyond the hype and hope: critical considerations for intranasal oxytocin research in autism spectrum disorder. Autism Research 2017, 10(1):25-41.

28. Knobloch HS, Charlet A, Hoffmann LC, Eliava M, Khrulev S, Cetin AH, Osten P, Schwarz MK, Seeburg PH, Stoop R: Evoked axonal oxytocin release in the central amygdala attenuates fear response. Neuron 2012, 73(3):553-566.

29. Oya K, Matsuda Y, Matsunaga S, Kishi T, Iwata N: Efficacy and safety of oxytocin augmentation therapy for schizophrenia: an updated systematic review and meta-analysis of randomized, placebo-controlled trials. European archives of psychiatry and clinical neuroscience 2016, 266(5):439-450.

30. Dodhia S, Hosanagar A, Fitzgerald DA, Labuschagne I, Wood AG, Nathan PJ, Phan KL: Modulation of resting-state amygdala-frontal functional connectivity by oxytocin in generalized social anxiety disorder. Neuropsychopharmacology 2014, 39(9):2061-2069.

---

## [Editor Report · Decision Letter 4]

17 Jan 2022

Large household reduces dementia mortality: A cross-sectional data analysis of 183 populations

PONE-D-21-26012R4

Dear Dr. Wenpeng You,

We’re pleased to inform you that your manuscript has been judged scientifically suitable for publication and will be formally accepted for publication once it meets all outstanding technical requirements.

Kind regards,

Wen-Wei Sung, M.D., Ph.D.

Academic Editor

PLOS ONE

---

## [Editor Report · Acceptance letter]

27 Jan 2022

PONE-D-21-26012R4 

Large household reduces dementia mortality: A cross-sectional data analysis of 183 populations 

Dear Dr. You:

I'm pleased to inform you that your manuscript has been deemed suitable for publication in PLOS ONE. Congratulations! Your manuscript is now with our production department. 

Kind regards, 

on behalf of

Dr. Wen-Wei Sung 

Academic Editor

PLOS ONE